# EAGLE-3: Scaling up Inference Acceleration of Large Language Models via Training-Time Test

**Yuhui Li[1,2], Fangyun Wei[3], Chao Zhang[2], Hongyang Zhang[1,4]**
[1]University of Waterloo [2]Peking University [3]Microsoft Research [4]Vector Institute
yuhui.li@stu.pku.edu.cn, fawe@microsoft.com
c.zhang@pku.edu.cn, hongyang.zhang@uwaterloo.ca

## Abstract

The sequential nature of modern LLMs makes them expensive and slow, and speculative sampling has proven to be an effective solution to this problem. Methods like EAGLE perform autoregression at the feature level, reusing top-layer features from the target model to achieve better results than vanilla speculative sampling. A growing trend in the LLM community is scaling up training data to improve model intelligence without increasing inference costs. However, we observe that scaling up data provides limited improvements for EAGLE. We identify that this limitation arises from EAGLE's feature prediction constraints. In this paper, we introduce EAGLE-3, which abandons feature prediction in favor of direct token prediction and replaces reliance on top-layer features with multi-layer feature fusion via a technique named training-time test. These improvements significantly enhance performance and enable the draft model to fully benefit from scaling up training data. Our experiments include both chat models and reasoning models, evaluated on five tasks. The results show that EAGLE-3 achieves a speedup ratio up to 6.5x, with about 1.4x improvement over EAGLE-2. In the SGLang framework, EAGLE-3 achieves a 1.38x throughput improvement at a batch size of 64. The code is available at https://github.com/SafeAILab/EAGLE.

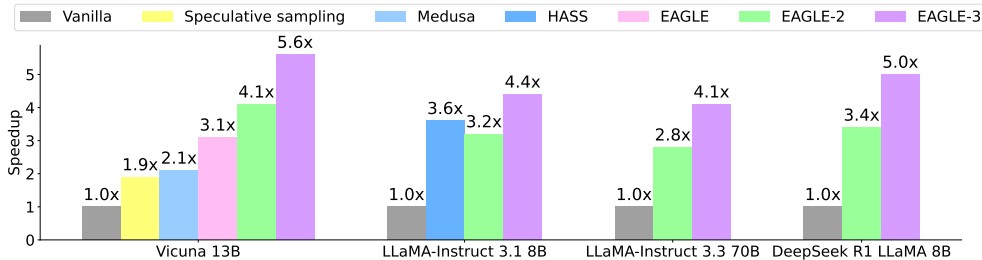

Figure 1: Speedup ratios of different methods at temperature=0. For the standard speculative sampling, Vicuna-13B uses Vicuna-68M as the draft model. In Table 1, we present comparisons with additional methods, but this figure only showcases a subset. Chat model's evaluation dataset is MT-bench, and the reasoning model's evaluation dataset is GSM8K. DeepSeek R1 LLaMA 8B refers to DeepSeek-R1-Distill-LLaMA 8B.

## 1 Introduction

Modern Large Language Models (LLMs) are being applied to more domains, with their improved capabilities driven by scaling model parameters—some LLMs now exceed hundreds of billions of

39th Conference on Neural Information Processing Systems (NeurIPS 2025).

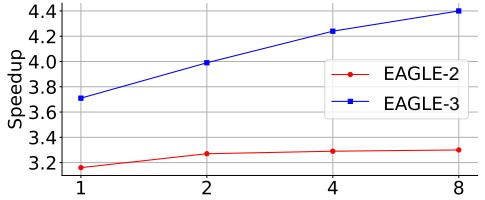 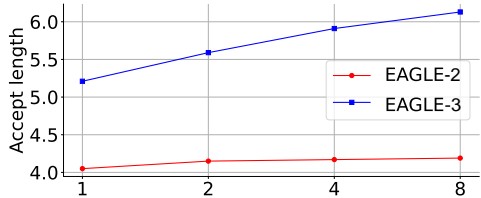

Figure 2: Scaling law evaluated on the MT-bench using LLaMA-Instruct 3.1 8B as the target model, with the x-axis representing the data scale relative to ShareGPT. The new architectural designs in EAGLE-3 enable an increasing scaling curve, which was never observed in the previous works.

parameters. In autoregressive generation, each token requires accessing all model parameters, making LLM inference slow and costly.

Recently, test-time scaling up has gained significant attention. Models like ChatGPT o1 and DeepSeek-R1 [1] engage in deliberate reasoning before responding, pushing the boundaries of LLM capabilities at the cost of longer inference time. However, these models often require lengthy reasoning processes, making them extremely costly, while the increased response time severely impacts user satisfaction. These reasoning models significantly increase the proportion of inference costs in the overall LLM pipeline, driving researchers to explore cheaper and faster inference optimization methods.

Speculative sampling methods can reduce LLM latency by partially parallelizing the generation process. These methods rapidly generate draft tokens and then verify them in parallel. This allows multiple tokens to be produced in a single forward pass, significantly reducing inference latency. As a state-of-the-art speculative sampling method, EAGLE [2] decodes an LLM by leveraging the top-layer features of the target model (i.e., the representations before the LM head) to perform *next-feature prediction*. As shown in the first box in Figure 3, EAGLE trains the draft model by feeding all previous features $f_1, ..., f_t$ into the target model to predict the next feature $\hat{f}_{t+1}$, with the training data feature $f_{t+1}$ as the label. In the test phase (the second box), EAGLE autoregressively predicts the next feature $\hat{f}$ and then uses the target model's LM head to obtain the draft token $\hat{t}$. By leveraging the rich information from the target model, EAGLE achieves significantly better acceleration compared to vanilla speculative sampling. Subsequent methods such as HASS [3] and Falcon [4] also adopt the approach of predicting the next feature using the current feature sequence.

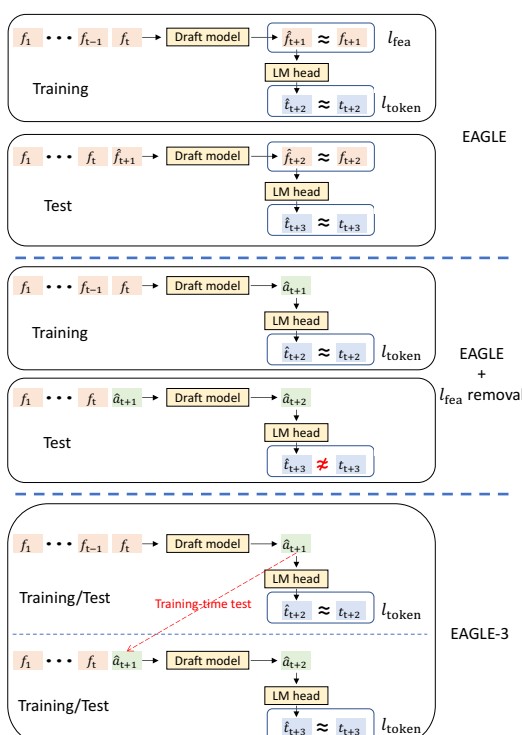

Figure 3: Illustration of **training-time test** (the bottom part) and its comparison with other draft methods (the upper and middle parts). $f$ denotes the feature, $t$ denotes the token, and $a$ represents the unconstrained vectors. We use the hat to denote the predictions from models. For EAGLE and EAGLE + $l_{fea}$ removal (the upper and middle parts), the training and test processes are different. However, for EAGLE-3 (the bottom part), the training and test processes are the same.

Recent LLMs have increasingly relied on larger training datasets to achieve better performance. For example, LLaMA series models with sizes of 7B (8B) have used 1T, 2T, and 15T tokens of training data for LLaMA 1 [5], LLaMA 2 [6], and LLaMA 3 [7], respectively, resulting in significant improvements across various metrics while keeping the model architecture and inference cost largely unchanged. Similarly, we aim to improve the acceptance rate and acceleration ratio of EAGLE by increasing its training data.

Unfortunately, we observe that the gains from additional training data for EAGLE are limited. We analyze the reasons behind this phenomenon. As shown in the upper part of Figure 3, in the test

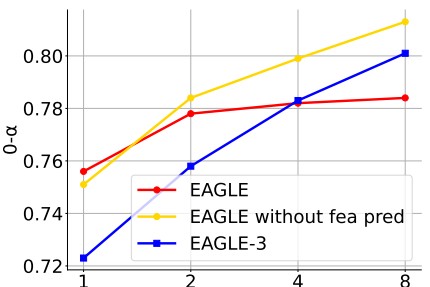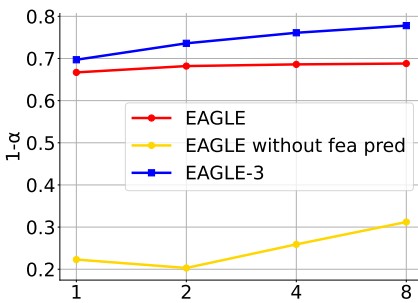

Figure 4: Comparison of acceptance rates across different methods, with the x-axis representing the data scale relative to ShareGPT.

phase, EAGLE performs autoregressive prediction at the feature level (as both input and output of the draft model are with hat), predicting the next feature and then feeding the feature into the LM head of the target model to obtain the token distribution. In the training phase, EAGLE's loss function consists of two components: the feature prediction loss $l_{\text{fea}}$ and the token prediction loss $l_{\text{token}}$. Thanks to the feature prediction loss, the draft model after training (when the input of the draft model is without hat and the output of the draft model is with hat) can adapt to the test case and acquire multi-step prediction capabilities. However, with token prediction as the ultimate goal, feature prediction can be seen as an additional constraint, which limits the expressiveness of the draft model and makes it difficult to benefit from increased data. After removing the feature constraint and expanding the training data (the middle part of Figure 3), as shown in Figure 4, the acceptance rate $0\text{-}\alpha$ of the first draft token improves significantly. However, the output of the draft model in the training phase, denoted as $\hat{a}_{t+1}$, is far away from the ground-truth $f_{t+1}$, causing the input sequence $f_1, f_2, \cdots, f_t, \hat{a}_{t+1}$ in the test phase to deviate significantly from the training distribution, resulting in a very low acceptance rate $1\text{-}\alpha$ for the second draft token, as shown in Figure 4. We can address this issue by moving the output $\hat{a}_{t+1}$ back into the input of draft model in the training process (the bottom of Figure 3), similar to autoregressive inference. Using this method, the benefits of increasing training data become more pronounced. We name this technique as training-time test.

EAGLE and speculative sampling methods such as Medusa [8] reuse the top-layer features of the target model, specifically the features immediately before the LM head. For an LM head with a full-rank weight matrix, the top-layer features corresponding to the logits of the next token are unique, ensuring that the information contained in these features aligns directly with the logits of the next token. However, predicting the next-next token based solely on top-layer features—which are inherently limited to the next token—poses a significant challenge. Fortunately, the training-time test technique described above enables the use of features from intermediate layers instead of relying solely on the top layer, as the feature prediction loss $l_{\text{fea}}$ has been removed during training.

This paper introduces EAGLE-3, an enhanced version of EAGLE that achieves a significant speedup:

- **A training-time test architecture for the draft model:** We remove the feature prediction constraint and directly predict tokens while simulating multi-step generation during training. This direct token prediction provides complete flexibility in the draft model's input. Instead of reusing only the top-layer features, we integrate and leverage low-, mid-, and high-level features from the target model, capturing rich semantic information from different layers.

- **A new scaling law for inference acceleration in LLMs:** With the new architecture, we observe that increasing the amount of training data for the draft model leads to a proportional increase in the speedup ratio of EAGLE-3. This scaling behavior was not observed in the original EAGLE architecture, as shown in Figure 2

- **Improved inference acceleration:** EAGLE-3, trained with approximately 8x more data than EAGLE, achieves a 1.4x latency speedup over EAGLE-2 at batch size 1. Speculative sampling is often thought to reduce throughput at large batch sizes. However, in SGLang [9], a production-grade framework, EAGLE-3 improves throughput by 38% at a batch size of 64. We expect larger data size would lead to further improved speedup ratio.

## 2 Preliminaries

### 2.1 Speculative Sampling

Speculative sampling [10, 11, 12, 13] is a lossless LLM acceleration technique that alternates between drafting and verification, where drafting is performed at low cost and verification is parallelized, corresponding to the generation of drafts and the verification process, respectively. We use $t_i$ to denote the $i$-th token and $T_{a:b}$ to represent the token sequence $t_a, t_{a+1}, \cdots, t_b$. When $T_{1:j}$ is used as the prefix, the two stages of speculative sampling are as follows.

In the drafting stage, speculative sampling utilizes a draft model (a smaller version from the same series as the target model) to autoregressively generate $k$ tokens to form the draft. $\hat{T}_{j+1:j+k}$, while also recording the probability $\hat{p}$ for each token.

In the verification stage, speculative sampling invokes the target model to evaluate the draft $\hat{T}_{j+1:j+k}$ and records its probability $p$. Speculative sampling then determines the acceptance of draft tokens sequentially, from front to back. For token $\hat{t}_{j+i}$, the probability of acceptance is given by $\min(1, p_{j+i}(\hat{t}_{j+i})/\hat{p}_{j+i}(\hat{t}_{j+i}))$. If the token is accepted, the process moves to the next token. Otherwise, a token is sampled from the distribution $\text{norm}(\max(0, p_{j+i} - \hat{p}_{j+i}))$ to replace $\hat{t}_{j+i}$, and the remaining tokens in the draft are discarded. Appendix A.1 of [10] proves that speculative sampling is consistent with the distribution of vanilla autoregressive decoding.

### 2.2 EAGLE and EAGLE-2

The draft model with limited capacity struggles to precisely approximate the large-scale target model. EAGLE leverages the top-layer features of the target model as additional information and performs autoregression at the feature level, simplifying the drafting process. EAGLE performs autoregression at the feature level and then uses the LM head of the target model to obtain the draft token. Due to the sampling results at the token layer being hidden, feature-level autoregression introduces uncertainty. EAGLE addresses this issue by feeding the token sequence from the previous time step, i.e., the sampling results, into the draft model. Unlike the chain-like drafts of Vanilla speculative sampling, EAGLE generates multiple draft tokens at the same position, resulting in a tree-like draft. In the verification stage, EAGLE uses tree attention to parallelize the verification of the draft tree. Interestingly, EAGLE inspired the *multi-token prediction* technique used in the pre-training of DeepSeek-v3 [14], which in turn inspired new architectural designs in EAGLE-3.

EAGLE [2] and Medusa [8], among others, use tree-shaped drafts, where the structure of the draft tree is predefined, static, and context-independent. The difficulty of drafting is closely related to the context, and a static draft tree can lead to resource wastage. EAGLE-2 [15] approximates the acceptance rate using the confidence of the draft model and dynamically generates the draft tree based on this, performing pruning of the draft tree at the end of the drafting stage. EAGLE-3 also adopts the context-aware dynamic draft tree proposed in EAGLE-2.

## 3 EAGLE-3

In this section, we provide a detailed description of the implementation of EAGLE-3.

### 3.1 Inference Pipeline

Consistent with other speculative sampling methods, EAGLE-3 alternates between the drafting and verification stages. The difference between EAGLE-3 and EAGLE lies in the drafting stage, which we introduce with an example, as shown in Figure 5. Consider the prefix "How can". During the prefill phase or the previous verification stage, the target model performs a forward pass to generate the next token, "I". We record the low, middle, and high-level feature sequences from the target model's forward pass, denoted as $l$, $m$, and $h$, respectively. We concatenate the $k$-dimensional vectors $l$, $m$, and $h$ to form a $3k$-dimensional vector, then pass it through a fully connected (FC) layer to reduce it to $k$-dimensions, obtaining a feature $g$ that integrates information from different layers. Here, $k$ refers to the hidden size of the target model.

Our goal is to generate a draft token sequence with the prefix "How can I". By inputting only $g_{\text{how}}$ and $g_{\text{can}}$, the draft model cannot access the random sampling process. Therefore, similar to EAGLE [2],

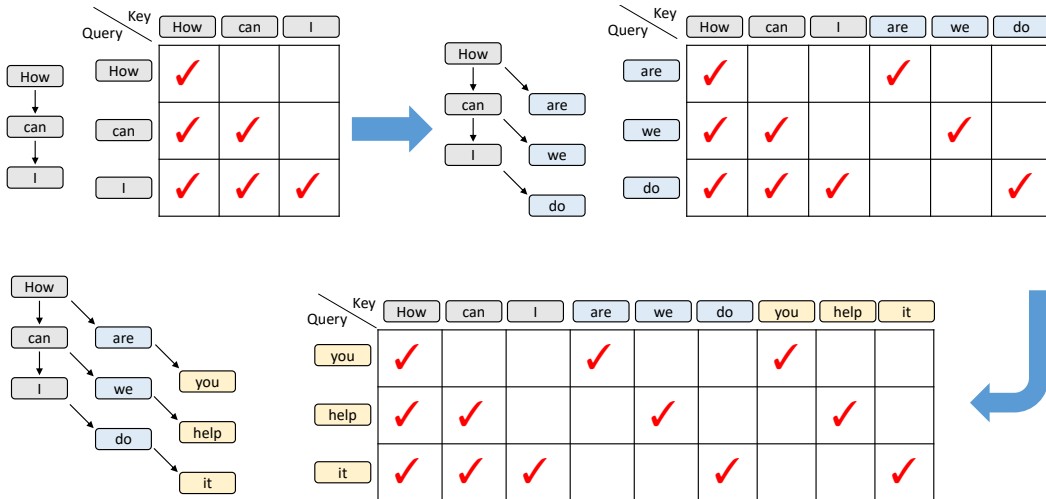

Figure 6: Diagram of the attention causal masks during training-time test. It sequentially shows a native training step (the first step) and two simulated training steps (the second and third steps). The arrows between tokens represent contextual relationships. The gray tokens represent the training data while the blue and yellow tokens represent the first- and second-round predictions by the draft model, respectively. We use the training dataset as the labels for each token position.

we introduce the embedding $e_I$ of the sampled token "I". The concatenated vector is then passed through an FC layer to reduce its dimensionality to $k$, and subsequently inputted into a single layer decoder, producing the output $a$. Finally, we input $a_I$ into the LM head and sample to obtain the draft token "do".

In Step 1, with the prefix "How can", we reuse $g_{how}$ and $g_{can}$ from the target model. In Step 2, the prefix becomes "How can I". Ideally, we would reuse $g_{how}$, $g_{can}$, and $g_I$ from the target model. However, this is not possible because the token "I" has not yet been checked by the target model, and we cannot obtain $g_I$. Instead, we use the output $a_I$ from the draft model in the previous step to replace $g_I$, and concatenate $a_I$ with the embedding $e_{do}$ of the sampled result "do" as the input to the draft model in Step 1. In Step 3, we similarly cannot obtain $g_{do}$, so we use $a_{do}$ as a replacement, concatenating $a_{do}$ with $e_{it}$ as the input to the draft model. The same approach is followed for subsequent steps.

## 3.2 Draft Model Training

The input to the draft model in EAGLE is either, or at least approximately, the top-layer features $f_1, f_2, \cdots, f_t$ of the target model. In contrast, the input to the draft model in EAGLE-3 may include the features $g_1, g_2, \cdots, g_t$ from the target model, or it may include the output $a_{t+1}, a_{t+2} \cdots, a_{t+j}$ from the draft model. Therefore, we need to train the draft model to adapt to different inputs. During training, we perform test steps, where we generate $a$ and feed it back into the draft model for further training.

The core of the draft model in EAGLE-3 is a Transformer decoder layer. Aside from the self-

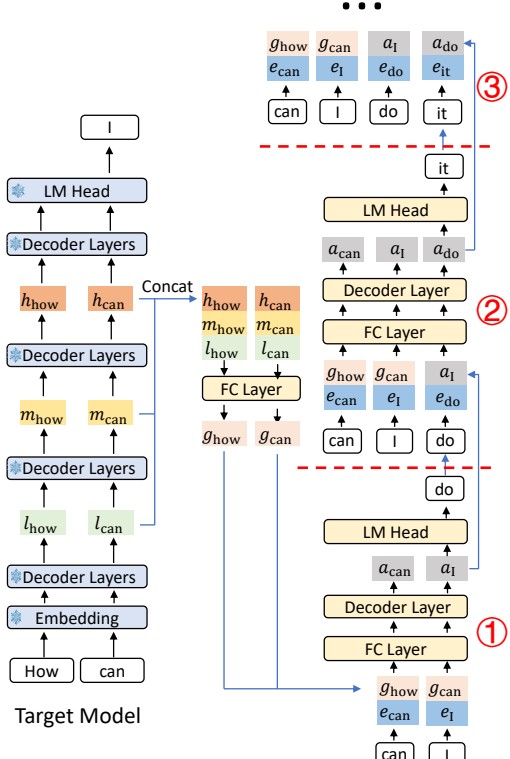

Figure 5: Diagram of the EAGLE-3 inference pipeline. $l$, $m$, and $h$ represent the low, middle, and high-level features of the target model, respectively. $e$ denotes the embedding.

attention operation, no other components interact with the context, so no further modifications are required during training or testing. The only component that requires slight modification is the self-attention, which we will describe in detail below.

Although the actual input consists of features, for clarity, we describe the process using tokens as input. As shown in Figure 6, the original training data is a sequence of length 3, "How can I", with a normal sequential dependency in the context. Therefore, the attention mask is a standard lower triangular matrix. The outputs at the three positions are "are", "we", and "do", which have a tree-like contextual relationship with "how", "can", and "I". As a result, when the input "are", "we", and "do" is fed into Step 2, the attention mask needs to be adjusted accordingly, as shown in the top-right corner of Figure 6. All attention masks are diagonal, except when the original training data is used as the key. Using matrix multiplication in this case would result in significant computational waste, so we can use vector dot products to calculate the attention score only for the corresponding positions.

HASS [3] and EAGLE-3 both make similar modifications to the attention mechanism to simulate the testing process during training, but this is not the main focus of EAGLE-3. The motivations, methods, and outcomes of the two approaches are distinctly different. The motivation behind HASS is to mitigate the error accumulation caused by inaccurate feature predictions in EAGLE. HASS still performs feature prediction, includes a feature prediction loss $l_{\text{fea}}$, and the input to the draft model must be the top-layer features. In contrast, the motivation behind EAGLE-3 is to remove unnecessary constraints to enhance the model's expressive power. EAGLE-3 no longer requires the draft model's output to fit the top-layer features of the target model, thus avoiding error accumulation. After removing feature prediction, the input to EAGLE-3 is completely free, and it is replaced by a fusion of features from different layers of semantic information. The removal of the feature prediction loss also enables us to discover a new scaling law for inference acceleration which was never found before. Figure 1 and Table 1 also shows the speedup of EAGLE-3 and HASS, with EAGLE-3 demonstrating significantly better performance.

## 4 Experiments

**Models.** We conduct experiments with open-source chat and reasoning models, including Vicuna 13B [16], LLaMA-Instruct 3.1 8B, LLaMA-Instruct 3.3 70B [7], and DeepSeek-R1-Distill-LLaMA 8B [17]. Due to the GPU constraint, we are unable to test EAGLE-3 on models larger than 70B.

**Draft Models.** Same to EAGLE and EAGLE-2, the draft model of EAGLE-3 consists of a single transformer layer. So, the scale of draft models in EAGLE, EAGLE-2, and EAGLE-3 is nearly the same.

**Tasks.** Following EAGLE [2] and Spec-Bench [18], we evaluate on five tasks, using the same weights for all tasks without fine-tuning on the respective tasks. For multi-turn conversation, code generation, mathematical reasoning, instruction following, and summarization,, we chose the MT-bench [19], HumanEval [20], GSM8K [21], Alpaca [22], and CNN/Daily Mail [23] datasets, respectively.

**Metrics.** EAGLE-3 does not modify the target model's weights and uses strict speculative sampling acceptance conditions, ensuring no loss in performance. Therefore, we do not evaluate generation quality. Instead, we use the following metrics to assess the acceleration performance:

- **Speedup Ratio:** The actual test speedup ratio relative to vanilla autoregressive decoding.

- **Average Acceptance Length $\tau$:** The average number of tokens generated per drafting-verification cycle, which corresponds to the number of tokens accepted from the draft.

- **Acceptance Rate $n$-$\alpha$:** The proportion of draft tokens accepted, which directly reflects the draft model's approximation to the target model. Following EAGLE's setup, we use a chain-like draft rather than a tree-like draft when testing acceptance rates. EAGLE suffers from error accumulation, meaning that the input to the draft model may be its own estimates rather than the exact values from the target model. Therefore, EAGLE uses $n$-$\alpha$ to represent the acceptance rate when the input contains $n$ estimated features, under the condition that the previous estimated tokens are all accepted by the target model. In other words, the acceptance rate for inputs $f_1, f_2, \cdots, f_i, \hat{f}_{i+1}, \cdots, \hat{f}_{i+n}$, where $f$ is the exact value and $\hat{f}$ is the draft model's estimate. Similarly, we use $n$-$\alpha$ to represent the acceptance rate in EAGLE-3 when the input contains $n$ self-predicted values $a$, i.e., the acceptance rate for inputs $g_1, g_2, \cdots, g_i, a_{i+1}, \cdots, a_{i+n}$, where $g$ is the fused feature from the target model.

Table 1: Speedup ratios and average acceptance lengths $\tau$ of different methods on A100 GPUs. V represents Vicuna, L31 represents LLaMA-Instruct 3.1, L33 represents LLaMA-Instruct 3.3, and DSL represents DeepSeek-R1-Distill-LLaMA. SpS denotes standard speculative sampling, with its draft model being Vicuna-68M. Methods like Medusa relax acceptance conditions under non-greedy settings, which do not guarantee lossless acceleration. Therefore, we do not compare EAGLE-3 with these methods when temperature=1.

| | | MT-bench | | HumanEval | | GSM8K | | Alpaca | | CNN/DM | | Mean | |
|---|---|---|---|---|---|---|---|---|---|---|---|---|---|
| Model | Method | Speedup | $\tau$ | Speedup | $\tau$ | Speedup | $\tau$ | Speedup | $\tau$ | Speedup | $\tau$ | Speedup | $\tau$ |
| | | Temperature=0 | | | | | | | | | | | |
| V 13B | SpS | 1.93x | 2.27 | 2.23x | 2.57 | 1.77x | 2.01 | 1.76x | 2.03 | 1.93x | 2.33 | 1.92x | 2.24 |
| | PLD | 1.58x | 1.63 | 1.85x | 1.93 | 1.68x | 1.73 | 1.16x | 1.19 | 2.42x | 2.50 | 1.74x | 1.80 |
| | Medusa | 2.07x | 2.59 | 2.50x | 2.78 | 2.23x | 2.64 | 2.08x | 2.45 | 1.71x | 2.09 | 2.12x | 2.51 |
| | Lookahead | 1.65x | 1.69 | 1.71x | 1.75 | 1.81x | 1.90 | 1.46x | 1.51 | 1.46x | 1.50 | 1.62x | 1.67 |
| | Hydra | 2.88x | 3.65 | 3.28x | 3.87 | 2.93x | 3.66 | 2.86x | 3.53 | 2.05x | 2.81 | 2.80x | 3.50 |
| | EAGLE | 3.07x | 3.98 | 3.58x | 4.39 | 3.08x | 3.97 | 3.03x | 3.95 | 2.49x | 3.52 | 3.05x | 3.96 |
| | EAGLE-2 | 4.26x | 4.83 | 4.96x | 5.41 | 4.22x | 4.79 | 4.25x | 4.89 | 3.40x | 4.21 | 4.22x | 4.83 |
| | EAGLE-3 | **5.58x** | **6.65** | **6.47x** | **7.54** | **5.32x** | **6.29** | **5.16x** | **6.17** | **5.01x** | **6.47** | **5.51x** | **6.62** |
| L31 8B | EAGLE-2 | 3.16x | 4.05 | 3.66x | 4.71 | 3.39x | 4.24 | 3.28x | 4.12 | 2.65x | 3.45 | 3.23x | 4.11 |
| | HASS | 3.55x | 4.41 | 3.78x | 4.85 | 3.45x | 4.47 | 3.57x | 4.55 | 2.77x | 3.55 | 3.42x | 4.37 |
| | EAGLE-3 | **4.40x** | **6.13** | **4.85x** | **6.74** | **4.48x** | **6.23** | **4.82x** | **6.70** | **3.65x** | **5.34** | **4.44x** | **6.23** |
| L33 70B | EAGLE-2 | 2.83x | 3.67 | 3.12x | 4.09 | 2.83x | 3.69 | 3.03x | 3.92 | 2.44x | 3.55 | 2.85x | 3.78 |
| | EAGLE-3 | **4.11x** | **5.63** | **4.79x** | **6.52** | **4.34x** | **6.15** | **4.30x** | **6.09** | **3.27x** | **5.02** | **4.12x** | **5.88** |
| DSL 8B | EAGLE-2 | 2.92x | 3.80 | 3.42x | 4.29 | 3.40x | 4.40 | 3.01x | 3.80 | 3.53x | 3.33 | 3.26x | 3.92 |
| | EAGLE-3 | **4.05x** | **5.58** | **4.59x** | **6.38** | **5.01x** | **6.93** | **3.65x** | **5.37** | **3.52x** | **4.92** | **4.16x** | **5.84** |
| | | Temperature=1 | | | | | | | | | | | |
| V 13B | SpS | 1.62x | 1.84 | 1.72x | 1.97 | 1.46x | 1.73 | 1.52x | 1.78 | 1.66x | 1.89 | 1.60x | 1.84 |
| | EAGLE | 2.32x | 3.20 | 2.65x | 3.63 | 2.57x | 3.60 | 2.45x | 3.57 | 2.23x | 3.26 | 2.44x | 3.45 |
| | EAGLE-2 | 3.80x | 4.40 | 4.22x | 4.89 | 3.77x | 4.41 | 3.78x | 4.37 | 3.25x | 3.97 | 3.76x | 4.41 |
| | EAGLE-3 | **4.57x** | **5.42** | **5.15x** | **6.22** | **4.71x** | **5.58** | **4.49x** | **5.39** | **4.33x** | **5.72** | **4.65x** | **5.67** |
| L31 8B | EAGLE-2 | 2.44x | 3.16 | 3.39x | 4.39 | 2.86x | 3.74 | 2.83x | 3.65 | 2.44x | 3.14 | 2.80x | 3.62 |
| | HASS | 2.58x | 3.31 | 3.48x | 4.50 | 2.87x | 3.77 | 3.04x | 3.98 | 2.42x | 3.11 | 2.89x | 3.73 |
| | EAGLE-3 | **3.07x** | **4.24** | **4.13x** | **5.82** | **3.32x** | **4.59** | **3.90x** | **5.56** | **2.99x** | **4.39** | **3.45x** | **4.92** |
| L33 70B | EAGLE-2 | 2.73x | 3.51 | 2.89x | 3.81 | 2.52x | 3.36 | 2.77x | 3.73 | 2.32x | 3.27 | 2.65x | 3.54 |
| | EAGLE-3 | **3.96x** | **5.45** | **4.36x** | **6.16** | **4.17x** | **5.95** | **4.14x** | **5.87** | **3.11x** | **4.88** | **3.95x** | **5.66** |
| DSL 8B | EAGLE-2 | 2.69x | 3.41 | 3.01x | 3.82 | 3.16x | 4.05 | 2.64x | 3.29 | 2.35x | 3.13 | 2.77x | 3.54 |
| | EAGLE-3 | **3.20x** | **4.49** | **3.77x** | **5.28** | **4.38x** | **6.10** | **3.16x** | **4.30** | **3.08x** | **4.27** | **3.52x** | **4.89** |

**Implementation.** We use the AdamW optimizer, with beta values $(\beta_1, \beta_2)$ set to (0.9, 0.95) and implemented gradient clipping of 0.5. The learning rate is set to 5e-5. We simulate 5 steps during training-time test. We use ShareGPT and UltraChat-200K [24] as training data, containing approximately 68K and 464K data entries, respectively. We call the target model to generate responses rather than using a fixed dataset. For the reasoning model DeepSeek-R1-Distill-LLaMA 8B, we also use the OpenThoughts-114k-math dataset for training. We use 16x A100 GPUs for the training of EAGLE-3 head for 70B models in two weeks. If not specified, we use the A100 GPU to test 70B models and the RTX 3090 for other models. The testing environment for all methods accelerating the same target model is identical.

**Comparison.** We use vanilla autoregressive decoding as the baseline, which serves as the benchmark for speedup ratios (1.00x). We compare EAGLE-3 with recent lossless speculative sampling methods, including standard speculative sampling [10, 11, 25], PLD [26], Medusa [8], Lookahead [27], Hydra [28], HASS [3], EAGLE [2], and EAGLE-2 [15].

## 4.1 Effectiveness

Figure 1 and Table 1 demonstrate the acceleration performance of EAGLE-3. On all tasks and target models, EAGLE-3 achieves the highest speedup ratio and average acceptance length. EAGLE-3 provides a speedup of approximately 3.0x-6.5x compared to vanilla autoregressive generation, with a 20%-40% improvement over EAGLE-2. Different tasks affect the draft model's acceptance rate, so both the average acceptance length and speedup ratio are task-dependent. Due to the presence of many fixed templates in code generation tasks, generating drafts is the easiest, which is why EAGLE-3 performs best on HumanEval, achieving a speedup ratio of up to 6.5x and an average acceptance length of up to 7.5. DeepSeek-R1-Distill-LLaMA 8B is an exception, with the highest

speedup ratio on the mathematical reasoning dataset GSM8K. This may be because we trained the draft model of DeepSeek-R1-Distill-LLaMA 8B using the OpenThoughts-114k-math dataset.

Figure 7 shows the acceptance rates of EAGLE and EAGLE-3 on MT-bench with LLaMA-Instruct 3.1 8B as the target model. The acceptance rate of EAGLE-3 is significantly higher than that of EAGLE. As the input from the draft model itself increases, the acceptance rate of EAGLE drops significantly, whereas EAGLE-3's acceptance rate remains almost unchanged, demonstrating the effectiveness of the Training-time test.

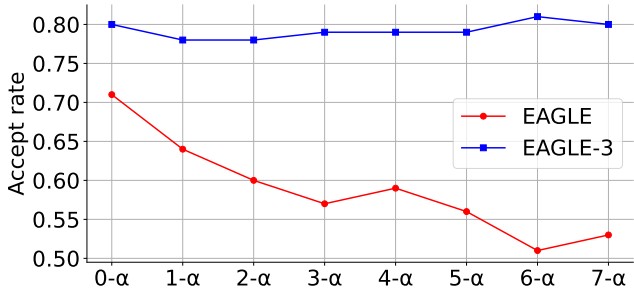

Figure 7: Acceptance rate of EAGLE and EAGLE-3 on MT-bench, with the target model being LLaMA-Instruct 3.1 8B. Hereby, $n$-$\alpha$ refers to the acceptance rate when the input contains $n$ estimated features, under the condition that the previous estimated tokens are all accepted by the target model.

## 4.2 Ablation Study

The improvements of EAGLE-3 mainly come from three aspects: first, the removal of the feature regression constraint, second, the improvement from reusing only the top-layer features to reusing a mix of low, middle, and high-level features, and third, the increase of training data. We conducted an ablation study on MT-bench with LLaMA-Instruct 3.1 8B as the target model. The results, shown in Table 2, indicate that the first and second improvements in EAGLE-3 significantly enhance the acceptance length and speedup ratio, demonstrating the rationality of the EAGLE-3 design. Figure 2 shows how the speedup ratio increases w.r.t. the amount of training data. The new architectural designs in EAGLE-3 enable an increasing scaling curve, which was never observed in the previous works.

Table 2: Ablation study results with LLaMA-Instruct 3.1 8B as the target model. "Remove fea con" refers to the first improvement of EAGLE-3, which removes the feature prediction constraint. "Fused features" refers to the second improvement of EAGLE-3, where low, middle, and high-level feature fusion replaces the use of top-layer features.

| Method | MT-bench | | GSM8K | |
|---|---|---|---|---|
| | Speedup | $\tau$ | Speedup | $\tau$ |
| EAGLE-2 | 3.16x | 4.05 | 3.39x | 4.24 |
| + remove fea con | 3.82x | 5.37 | 3.77x | 5.22 |
| + fused features (ours) | 4.40x | 6.13 | 4.48x | 6.23 |

## 4.3 EAGLE-3 in SGLang

Speculative sampling algorithms reduce memory accesses and lower latency during memory-bound decoding by leveraging redundant computational power. As batch sizes increase, this redundancy decreases, reducing the effectiveness of speculative sampling. Efficiency improvements are more challenging in highly optimized production-grade frameworks. The performance of EAGLE-3 for large batches on a single H100 GPU and LLaMA-Instruct 3.1 8B in the SGLang v0.4.4 environment [9] was evaluated in Table 3. This part of the experiment did not use the tree structure, the chain length was set to 3, and the testing dataset was MT-Bench. EAGLE reduces throughput at batch size of 24, whereas EAGLE-3 still achieves a 38% throughput improvement at a batch size of 64.

Table 3: Throughput improvement under different batch sizes on H100 and LLaMA-Instruct 3.1 8B for the MT-Bench dataset, with SGLang without speculative sampling as the baseline (1.00x).

| Batch size | 2 | 4 | 8 | 16 | 24 | 32 | 48 | 56 | 64 |
|---|---|---|---|---|---|---|---|---|---|
| EAGLE | 1.40x | 1.38x | 1.23x | 1.02x | 0.93x | 0.94x | 0.88x | 0.99x | 0.99x |
| EAGLE-3 | 1.81x | 1.82x | 1.62x | 1.48x | 1.39x | 1.32x | 1.38x | 1.34x | 1.38x |

We also tested the throughput of EAGLE-3 at batch size = 1 on H100 when the target model is LLaMA-Instruct 3.1 8B and the testing dataset is MT-bench. The results are shown in Table 4.

Table 4: Throughput at batch size = 1 on a single H100 GPU when the target model is LLaMA-Instruct 3.1 8B and the testing dataset is MT-bench.

| Method | Throughput (bs=1) |
|---|---|
| SGLang (w/o speculative, 1x H100) | 158.34 tokens/s |
| SGLang + EAGLE-2 (1x H100) | 244.10 tokens/s |
| SGLang + EAGLE-3 (1x H100) | 373.25 tokens/s |

## 4.4 EAGLE-3 in vLLM

We also conducted a study on the impact of EAGLE-3 on throughput for large batch sizes based on vLLM [29], a widely used production-grade framework, and the results on RTX3090 and LLaMA-Instruct 3.1 8B are shown in Table 5. EAGLE shows the maximum throughput improvement at a batch size of 24, while EAGLE-3 shows this at 56. This part of the experiment did not use the tree structure, the maximum chain length was set to 2, and the testing dataset was MT-Bench.

Table 5: Throughput improvement under different batch sizes on RTX3090 and LLaMA-Instruct 3.1 8B for the MT-Bench dataset, with vLLM without speculative sampling as the baseline (1.00x).

| Batch size | 2 | 4 | 8 | 16 | 24 | 32 | 48 | 56 |
|---|---|---|---|---|---|---|---|---|
| EAGLE | 1.30x | 1.25x | 1.21x | 1.10x | 1.03x | 0.93x | 0.82x | 0.71x |
| EAGLE-3 | 1.75x | 1.68x | 1.58x | 1.49x | 1.42x | 1.36x | 1.21x | 1.01x |

## 5 Related Work

Many methods have been used to accelerate inference in LLMs, such as quantization [30, 31, 32, 33, 34] and distillation [35]. These methods generally have trade-offs, where there is a need to balance model performance with acceleration benefits.

Speculative sampling uses the target model for verification to ensure lossless acceleration. Early speculative decoding methods [36, 37] accelerated generation in greedy settings, while [10, 11] introduced speculative sampling to extend the draft verification framework to non-greedy generation. Many subsequent works have improved upon speculative sampling. EAGLE [2], EAGLE-2 [15], Medusa [8], and Hydra [28] reused the features of the target model. Based on the framework of EAGLE, HASS [3] simulates a multistep draft process during training to mitigate the issues of training-inference inconsistency and error accumulation in EAGLE. GLIDE and CAPE [38] reuse the target model's KV cache, while methods [39, 40, 41, 42, 43, 44, 45, 46, 47] like Draft & Verify [48] use layer skipping or early exits to reuse parts of the target model's parameters.

There are several key differences between HASS [3] and EAGLE-3. First, HASS drafts using only top-layer features, whereas EAGLE-3 integrates low-, mid-, and high-level features. Second, HASS retains the token loss $l_{token}$, while EAGLE-3 removes it to improve model capacity. Third, unlike HASS, EAGLE-3 exhibits a clear scaling law trend. Finally, EAGLE-3 significantly outperforms HASS, as demonstrated in Figure 1 and Table 1.

# 6 Conclusion

In this paper, we introduce EAGLE-3. Building upon EAGLE, EAGLE-3 incorporates two key improvements. First, it removes the feature prediction constraint, instead directly predicting draft tokens through a Training-time test. Second, it replaces the use of the target model's top-layer features with a fusion of the target model's lower, middle, and upper-layer features to obtain richer information. With these improvements, EAGLE-3 continues to benefit from the augmentation of training data, achieving a maximum speedup of 6.5x.

## Acknowledgement

We would like to thank James Liu, Ke Bao, Yineng Zhang, Lianmin Zheng, Ying Sheng, and many others in the SGLang team for evaluating EAGLE-3 in the SGLang environment. Hongyang Zhang is supported by the NSERC Discovery Grant RGPIN-2022-03215, DGECR-2022-00357, and Google Research Scholar Award.

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

# A  Implementation Details

**Vanilla:** We use models from the Huggingface.transformers library with the PyTorch backend and pre-allocated KV cache. Other methods also use these models as their base.

**(Standard) Speculative Sampling:** We use the assisted generation feature from the HuggingFace Transformers library.

**PLD, Lookahead, Medusa, and Hydra:** We use the default settings and the officially released weights.

**EAGLE:** Vicuna and LLaMA2-Chat draft models use the officially released weights, while LLaMA3-Instruct is trained using the ShareGPT dataset (consistent with Medusa and Hydra).

**EAGLE-2:** For the 7B (8B), 13B, and 70B original LLMs, we set the total number of draft tokens to 60, 50, and 48, respectively, with a draft tree depth of 6, and select 10 nodes during the expansion phase.

**EAGLE-3:** EAGLE-3's draft model achieves a significantly higher acceptance rate, allowing us to increase the draft tree depth from 6 to 8 while keeping the number of nodes the same as in EAGLE-2.

# B  A Comparative Study of EAGLE-3 and HASS

The work most similar to EAGLE-3 is HASS. Both approaches simulate multi-step prediction during training, but this is neither the main focus of EAGLE-3 nor HASS. Training-time testing primarily involves adjusting the attention mask to enforce correct dependencies, which essentially simplifies tree attention into a fixed-shape form (as illustrated in Figure 6). In fact, tree attention has been widely adopted in nearly all speculative decoding methods proposed in recent years. Feeding model outputs instead of ground truth during training, known as scheduled sampling, was also widely explored in the RNN era.

The core contribution of HASS lies in identifying the train-test mismatch in EAGLE and mitigating it through tree attention-based simulation. In contrast, EAGLE-3 focuses on a different issue: the inability of EAGLE to benefit from data scaling. EAGLE-3 attributes this limitation to the feature prediction constraint—an issue also present in HASS. EAGLE-3 removes this constraint and uses tree attention for simulation. This modification enables EAGLE-3 to scale effectively with increased training data, whereas HASS does not. The ability to scale with data is the core contribution of EAGLE-3.

Figure 8 illustrates the performance of EAGLE-3 and HASS across different training data scales. Similar to EAGLE-2, HASS fails to scale up, while EAGLE-3 exhibits rapid performance improvements as more training data becomes available. Moreover, EAGLE-3 identifies that the top-layer features (used in EAGLE and subsequent works including HASS) tend to overfit to next-token prediction and are not well-suited for multi-step draft generation. To address this, EAGLE-3 replaces the top-layer features with a fusion of multi-level features. Therefore, EAGLE-3 also outperforms HASS when trained on smaller datasets.

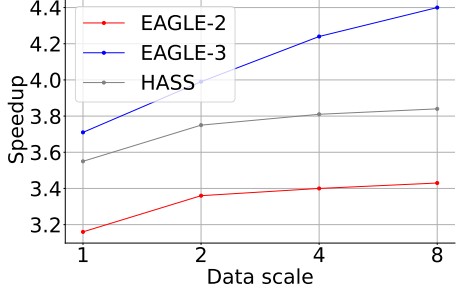 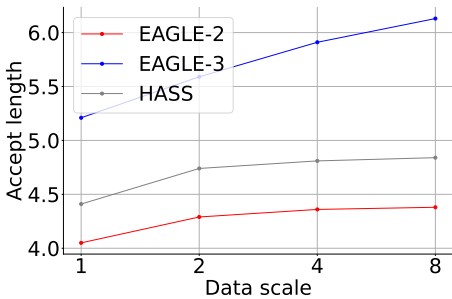

Figure 8: Scaling law evaluated on the MT-bench using LLaMA-Instruct 3.1 8B as the target model, with the x-axis representing the data scale relative to ShareGPT.

