# OpenReview forum: "EAGLE-3: Scaling up Inference Acceleration of Large Language Models via Training-Time Test"
_NeurIPS.cc/2025/Conference — NeurIPS 2025 poster_

### Official Review · Reviewer_szc3 · 2025-06-16

**Clarity:** 3
**Significance:** 3
**Originality:** 3
**Rating:** 4
**Confidence:** 3

**Summary:**

This paper proposed a new speculative sampling framework, named EAGLE3, which improves the performance of the draft model by increasing the training data quality and quantity, reusing the top/middle/bottom features for fusing, and removing the feature constrained. Experimental results demonstrated that EAGLE3 improves the inference speed compared to the baselines across various framework.

**Questions:**

See weakness above.

**Ethical Concerns:**

["NO or VERY MINOR ethics concerns only"]

**Final Justification:**

The authors' rebuttal has addressed my previous concerns. I will maintain my positive score.

**Limitations:**

yes

**Paper Formatting Concerns:**

No.

**Quality:**

3

**Strengths And Weaknesses:**

Strength
1. The motivation of EAGLE3 is clear , speeding up the LLM inference is a fundamental problem in practice.
2. The design of EAGLE3 is generally clear and innovative.
3. The experimental results are solid. The performance of EAGLE3 are evaluated across various inference frameworks like siglang and vllm.

Weakness
1. As mentioned in the paper, “DeepSeek-v3 inspired the design of EAGLE3”, some discussions and comparisons between the design of deepseek-v3 and that of EAGLE3 could be added to the paper.
2. The ablation study showed that a mix of low, middle, and high-level features improves the performance, is there any insights or discussions on why the fused features can improve the performance of the draft model? As in deepseek-v3, only the top features is taken as the input of the multi-token prediction decoder, it is recommended to add some discussions on this point.
3. In Table 3-5, only 2 out of three algorithms (EAGLE1, EAGLE2, and EAGLE3) are evaluated, it is recommended to evaluate all of them.

---

> ### Author Rebuttal · Authors · 2025-07-30
>
> Thank you for the thorough evaluation and insightful suggestions, which have been instrumental in refining our work.
>
> > Q1. As mentioned in the paper, “DeepSeek-v3 inspired the design of EAGLE3”, some discussions and comparisons between the design of deepseek-v3 and that of EAGLE3 could be added to the paper.
>
>
> A1. The Multi-Token Prediction (MTP) architecture in DeepSeek-V3 shares similarities with EAGLE. The key difference lies in that EAGLE reuses a single network in a loop to predict all future tokens, which requires a feature loss to align the input and output features. In contrast, DeepSeek-V3's MTP uses multiple separate networks to predict tokens at different positions, thereby avoiding the need for feature loss at the cost of increased computational overhead.
>
> This design inspired us to remove the feature loss in EAGLE-3 (similar to DeepSeek-V3), while still using a single network (as in EAGLE), thanks to the simulation mechanism during training.
>
> > Q2. The ablation study showed that a mix of low, middle, and high-level features improves the performance, is there any insights or discussions on why the fused features can improve the performance of the draft model? As in deepseek-v3, only the top features is taken as the input of the multi-token prediction decoder, it is recommended to add some discussions on this point.
>
>
> A2. We discussed this point in the original manuscript, Lines 83–88. The original text is as follows:
>
> EAGLE and speculative sampling methods such as Medusa reuse the top-layer features of the target model, specifically the features immediately before the LM head. For an LM head with a full-rank weight matrix, the top-layer features corresponding to the logits of the next token are unique, ensuring that the information contained in these features aligns directly with the logits of the next token. However, predicting the next-next token based solely on top-layer features—which are inherently limited to the next token—poses a significant challenge.
>
> > Q3. In Table 3-5, only 2 out of three algorithms (EAGLE1, EAGLE2, and EAGLE3) are evaluated, it is recommended to evaluate all of them.
>
> A3. The difference between EAGLE-1 and EAGLE-2 lies in the draft tree generation process. However, the high batch size settings used in Tables 3 and 5 are not well-suited for tree-based drafting. This is because tree-based drafting is more computationally intensive, and the redundant GPU resources required for such speculative strategies are scarce under high batch size configurations. As a result, both methods employ chain-based drafts in these experiments, rendering EAGLE-1 and EAGLE-2 functionally equivalent under this setting. Regarding other methods, since SGLang only supports the EAGLE series, we did not include additional baselines to ensure a fair comparison.
>
> We provide additional experimental results for Table 4 below.
>
> | Method | SGLang    | SGLang+EAGLE-1    | SGLang+EAGLE-2 | SGLang+EAGLE-3|
> |------------|------|------|------|------|
> | Throughput    | 158.34tokens/s  |219.18tokens/s  | 244.10tokens/s | 373.25tokens/s |

---

> > ### Comment · Reviewer_szc3 · 2025-08-05
> >
> > Thanks for the authors' rebuttal, which has addressed my previous concerns. Please also incorporate the clarifications and the additional experiment in the revision. I will maintain my positive score.

---

### Official Review · Reviewer_9fyH · 2025-07-01

**Clarity:** 2
**Significance:** 3
**Originality:** 3
**Rating:** 4
**Confidence:** 4

**Summary:**

This paper proposes EAGLE-3, a new speculative decoding framework for accelerating large language model (LLM) inference. The main contributions are: (1) Compared to previous EAGLE series, EAGLE-3 removes the feature prediction loss and retains only the token prediction loss, and introduces a training-time test mechanism so that the draft model learns to generate token sequences more likely to be accepted by the main model during training. (2) EAGLE-3 introduces multi-layer feature fusion, concatenating low-, mid-, and high-level hidden states and fusing them via a fully connected layer for draft model token prediction. Experiments on LLaMA, Vicuna, and other mainstream LLMs show up to 6.5x speedup without loss of generation quality, and demonstrate improved scaling law behavior compared to EAGLE-2.

**Questions:**

- **Can you provide a clear mathematical formulation for the multi-layer feature fusion mechanism?** Section 3.1 only gives a high-level description. Please specify the exact formula and rationale for the choice of layers.
- **How are the low-, mid-, and high-level features selected?** Please clarify which layers are used and provide ablation studies for different layer combinations, not just with/without fusion (see Table 2).
- **Can you provide a theoretical analysis for why multi-layer feature fusion improves draft model prediction?** This is important for understanding the generality and limitations of the approach.
- **Is there any theoretical justification for the effectiveness of the training-time test mechanism on draft model generalization?** Please discuss or analyze its impact beyond empirical results.
- **Can you provide controlled experiments where EAGLE-3 and baselines (e.g., HASS ) are trained with the same amount of data?** This would clarify the contribution of the method versus data scale.
- **Do you plan to evaluate EAGLE-3 on larger or more diverse models (e.g., Qwen-32B or MoE)?** Please discuss the potential challenges and applicability. Also, will the scaling law also works on larger or more diverse models?

**Ethical Concerns:**

["NO or VERY MINOR ethics concerns only"]

**Final Justification:**

Thank you for the thorough rebuttal. I appreciate the effort you've put into addressing my concerns, but I'm still not convinced by the responses to Q1/Q2 and therefore will maintain my original score.
For Q1, while you provide mathematical formulas and ablation studies, the layer selection strategy lacks theoretical foundation. The fixed rule (3rd, n//2-th, n-3-th layers) appears to be empirically driven without explaining why these specific layers capture the most relevant information for speculative decoding. More critically, applying the same layer selection criteria across all model architectures is questionable—different models with varying depths and architectural designs may require adaptive strategies rather than a one-size-fits-all approach. The absence of cross-architectural validation and semantic analysis of selected layers weakens the scientific rigor of this core component.
For Q2, the evaluation on larger models remains insufficient. Testing only LLaMA4 variants doesn't demonstrate generalizability across diverse architectures (Qwen, Gemma, etc.). The acknowledged incomparability between SGLang and Huggingface results further undermines the evidence. Additionally, the notably lower speedups on MoE models (2.18x vs 4.40x on dense models) without analysis or explanation raises concerns about the method's scalability and robustness across different model types.

**Limitations:**

The authors discuss some limitations (Section 6), but do not sufficiently address the lack of theoretical analysis, the limited model coverage, or the fairness of experimental comparisons. I encourage the authors to be more explicit and thorough in discussing these points.

**Quality:**

2

**Strengths And Weaknesses:**

**Strengths:**

- The method is novel and the experimental evaluation is extensive, covering multiple LLMs and tasks. The results are convincing and show significant speedup.
- LLM inference acceleration is a highly impactful problem for both academia and industry. EAGLE-3 demonstrates strong practical value.
- EAGLE-3 makes clear advances over the EAGLE series by removing feature prediction loss, introducing training-time test, and using multi-layer feature fusion. These design choices lead to better scaling and draft model performance.
- The paper is well organized and generally easy to follow.

**Weaknesses:**

- The paper lacks theoretical analysis for its core innovations. Multi-layer feature fusion and training-time test are only motivated intuitively, with no mathematical formulation and theoretical justification. This is a major weakness for NeurIPS standards.
- The implementation of multi-layer feature fusion is described only at a high level, with no clear mathematical formula and rationale for layer selection. The ablation study only compares with/without fusion, lacking analysis of different layer combinations.
- Experiments are limited to LLaMA, Vicuna, and similar models (up to 70B). There is no evaluation on more recent or larger models (e.g., Qwen-32B or MoE), which limits claims of generality.
- EAGLE-3 is trained with more data than baselines, which may confound the effect of the method with data scale. There is no controlled comparison under equal data conditions.
- The discussion of limitations is brief and does not address the above theoretical and practical issues in depth.

---

> ### Author Rebuttal · Authors · 2025-07-30
>
> Thank you for your valuable suggestions. In what follows, I will address your questions and clarify some misunderstandings.
>
> > Q1. The implementation of multi-layer feature fusion is described only at a high level, with no clear mathematical formula and rationale for layer selection. The ablation study only compares with/without fusion, lacking analysis of different layer combinations.
>
>
>
> A1. We select the 3rd, $n//2$-th, and $(n{-}3)$-th layers, where $n$ is the total number of layers. Let the corresponding feature vectors be denoted as $l$, $m$, and $h$, respectively. The fused feature is computed as $W \cdot \text{concat}(l, m, h)$, where $\text{concat}(\cdot)$ denotes vector concatenation, and $W \in \mathbb{R}^{h \times 3h}$ is a learnable projection matrix, with $h$ being the hidden dimension of the target model. The following is an ablation study on layer selection. As long as the final layer is not used and the features are fused, the performance improves significantly, while the specific choice of layers has limited impact. Furthermore, increasing the number of layers beyond three does not bring additional gains. Therefore, we opt for the $3$-rd, $n//2$-th, and $(n{-}3)$-th layers in our design.
>
> | Layers | Accept Length    | Speedup    |
> |------------|------|------|
> | $n$      | 5.37 | 3.82x |
> | $n-1$    | 5.65 | 4.00x |
> | $n-2$    | 5.76 | 4.06× |
> | $n-3$    | 5.76 | 4.05x |
> | $n-3$, $n−3$    | 5.93 | 4.26x |
> | $n-3$, $n//2$    | 6.02 | 4.31x |
> | $3$, $n//2$, $n−3$   | 6.13 | 4.40× |
> | $1$, $n//2$, $n−1$   | 6.11 | 4.38× |
> | $5$, $n//2$, $n−5$   | 6.09 | 4.37× |
> | $3$, $n//2-5$, $n//2+5$, $n−3$   | 6.11 | 4.37× |
> | $3$, $10$, $n//2$, $n-10$, $n−3$   | 6.15 | 4.39× |
>
> > Q2. Experiments are limited to LLaMA, Vicuna, and similar models (up to 70B). There is no evaluation on more recent or larger models (e.g., Qwen-32B or MoE), which limits claims of generality.
>
>
> A2. We provide additional results on larger MoE model architectures. The table below shows the acceleration performance of EAGLE-3 within the SGLang framework on LLaMA4 Scout (MoE, 109B) and LLaMA4 Maverick (MoE, 400B). It is important to note that results obtained under the production-grade SGLang framework should not be directly compared with those from standard Huggingface-based environments.
>
> | Model | Llama 4 Scout (MoE, 109B)   | Llama 4 Maverick (MoE, 400B)   |
> |------------|------|------|
> | Speedup      | 2.00x | 2.18x |
>
> > Q3. EAGLE-3 is trained with more data than baselines, which may confound the effect of the method with data scale. There is no controlled comparison under equal data conditions.
>
> A3. Figure 2 compares EAGLE-2 and EAGLE-3 under the same data scale. Futhermore, we have already included a comparison under the same data conditions with HASS—the most similar and strongest-performing baseline to our method—in Appendix B of the original manuscript. Similarly to EAGLE-2, HASS fails to scale up, while EAGLE-3 exhibits rapid performance improvements as more training data become available. EAGLE-3 consistently outperforms HASS across all data scales. For convenience, we paste the results here. Note that Data Scale = 1 refers to using the full amount of ShareGPT data. The evaluation is conducted on the MT-Bench dataset, with LLaMA-Instruct 3.1 8B as the target model.
>
> | Data Scale | 1    | 2    | 4 | 8|
> |------------|------|------|------|------|
> | EAGLE-2    | 3.16x  |3.36x  | 3.40x | 3.43x |
> | HASS   | 3.55x | 3.75x | 3.81x | 3.84x |
> | EAGGLE-3   | 3.71x | 3.99x | 4.24x | 4.40x |

---

> ### Author Response · Authors · 2025-08-05
>
> Dear Reviewer 9fyH,
>
> Thank you once again for your thoughtful and constructive feedback on our paper. We truly appreciate the time and effort you’ve dedicated to the review process.
>
> As a gentle reminder, the discussion period will conclude on August 6 (AoE). We’ve provided detailed responses to your comments in our rebuttal and sincerely hope they address your concerns.
>
> If you have any further questions or additional suggestions, we’d be more than happy to address them.
>
> Thank you again for your support and engagement!
>
> Warm regards,
>
> Authors of Submission 13079

---

### Official Review · Reviewer_YVao · 2025-07-03

**Clarity:** 3
**Significance:** 4
**Originality:** 3
**Rating:** 5
**Confidence:** 2

**Summary:**

This work presents EAGLE-3, which introduces a novel training strategy called training-time test, where the draft model simulates multi-step generation during training. This improves alignment between training and inference. Unlike the original EAGLE that predicts the next feature, EAGLE-3 performs token-level drafting, directly predicting tokens. This eliminates the need for feature prediction loss and allows for more expressive draft models. It also employs multi-layer feature fusion, leveraging features from low, middle, and high layers of the target model to provide richer semantic context. Moreover, EAGLE-3 is scalable with more training data for speculative decoding.

In the benchmark, EAGLE-3 achieves up to 6.5× speedup over standard autoregressive decoding and is 1.4× faster than EAGLE-2. When deployed in SGLang, it delivers a 1.38× throughput improvement at batch size 64, highlighting its practical efficiency. This is an impactful and strong work with open source project integration.

**Questions:**

none

**Ethical Concerns:**

["NO or VERY MINOR ethics concerns only"]

**Final Justification:**

Concerns addressed

**Limitations:**

yes

**Paper Formatting Concerns:**

it was pointed out by reviewer o9ZR02. Otherwise nothing major.

**Quality:**

3

**Strengths And Weaknesses:**

strength:
1. address an important question - we always want to reduce the cost of inference of LLMs
2. significant inference speedup and production impact
 - achieves up to 6.5x speedup, outperforming all previous speculative sampling methods including EAGLE-2, HASS, and Medusa.
 - evaluated in SGLang and vLLM, EAGLE-3 delivers real throughput improvements (e.g., 1.38x at batch size 64), showing real-world viability.
3. data scaling behavior
 - the paper introduces a scaling law: as training data increases, performance continues to improve, which was absent in previous spec decode work.
4. Training-time test strategy
 - Aligns the training and inference pipelines by simulating multi-step generation during training.
 - Reduces distribution shift between training and deployment, mitigating error accumulation. It's done via removing feature level predicition, and fusing low/mid/high level features to predict tne token

weakness:
1. limited model architecture coverage, MoE is becoming the mainstream backbone and getting more important
2. data scaling means that training the draft model becomes more expensive (but if the inference cost still dominates, that's fine. It's just not friendly to labs)
3. limited applicability for proprietary models where internal features cannot be accessed, such as openai gpt or claude models

---

> ### Author Rebuttal · Authors · 2025-07-30
>
> Thank you for your thoughtful review and constructive feedback. We truly value your input and have addressed your concerns in the following responses.
>
> > Q1. Limited model architecture coverage, MoE is becoming the mainstream backbone and getting more important.
>
>
> A1. EAGLE-3 is also applicable to accelerating MoE models. The table below shows the acceleration performance on LLaMA4 Scout (MoE, 109B) and LLaMA4 Maverick (MoE, 400B) within the SGLang framework. It is important to note that results obtained in the production-grade SGLang framework should not be directly compared with those from standard Huggingface-based setups.
>
> | Model | Llama 4 Scout (MoE, 109B)   | Llama 4 Maverick (MoE, 400B)   |
> |------------|------|------|
> | Speedup      | 2.00x | 2.18x |
>
> > Q2. Limited applicability for proprietary models where internal features cannot be accessed, such as openai gpt or claude models.
>
> A2. OpenAI and Anthropic can train EAGLE-3 heads for their proprietary models.
> For those models, EAGLE-3 can reduce deployment costs for companies by improving generation speed. This, in turn, lowers the cost for end users and enhances the overall user experience.

---

> > ### Comment · Reviewer_YVao · 2025-08-04
> >
> > Thank you for the rebuttal. I’ll keep the score.

---

> > > ### Author Response · Authors · 2025-08-05
> > >
> > > Dear Reviewer YVao,
> > >
> > > Thank you for your support and constructive comments.
> > >
> > > Warm regards,
> > >
> > > Authors of Submission 13079

---

### Official Review · Reviewer_o9ZR · 2025-07-03

**Clarity:** 3
**Significance:** 3
**Originality:** 3
**Rating:** 5
**Confidence:** 3

**Summary:**

In this manuscript, the authors propose a novel speculative decoding approach named EAGLE-3. The draft model performs autoregressive generation within an independent latent state space, enabling fast and accurate prediction of subsequent tokens. To enhance the informativeness of these predictions, the method incorporates hidden state fusion, allowing the draft model to access richer contextual representations. This combination endows the draft model with greater potential capacity, which is further unlocked by scaling up training. The proposed method is evaluated across various models, achieving promising results. Finally, the authors port their approach to different inference frameworks to examine its potential under batched decoding scenarios.

**Questions:**

1. Could the authors provide detailed specifications regarding the model and training configurations used in the study?

2. Could the authors explain the rationale behind the specific design choices adopted in the inference framework? Additionally, it would be helpful to further elaborate on the compatibility of EAGLE-3 with these deployment frameworks.

3. While training time may not be the most critical factor, the reported two-week training using 16 A100 GPUs appears to raise the practical barrier to adoption. Could the authors offer further explanation or justification for this level of computational demand?

**Ethical Concerns:**

["NO or VERY MINOR ethics concerns only"]

**Final Justification:**

**Addressed Issues**
- The authors adequately addressed the first concern by providing additional experiments and clarifications.
- For the second concern, the authors provided a thorough and well-reasoned analysis that helps clarify the root cause, even though a complete solution has not yet been achieved.

**Limitations:**

yes

**Paper Formatting Concerns:**

Figure 7 overlaps with the preceding paragraph

**Quality:**

3

**Strengths And Weaknesses:**

**Strengths:**

1. The proposed fusion of hidden state information and the removal of the constraint loss significantly enhance the learning capacity of the draft model. This provides new insights into how to effectively train draft models.

2. The authors conduct training across models of various sizes, demonstrating the robustness and scalability of EAGLE-3 under different base model configurations.

3. Despite space limitations, the authors port their implementation to practical deployment frameworks and discuss the acceleration potential under realistic conditions. This brings the evaluation closer to production-level settings.

**Weaknesses:**

1. While the paper mentions the fusion of three hidden layers (denoted as $l$, $m$, and $h$), it provides no details on how these specific layers are selected. Nor does it discuss how different choices might impact performance, such as whether three layers are optimal compared to using two or four. This lack of clarity obscures the model architecture and leaves readers without sufficient information or confidence to reproduce the reported results.

2. The authors do not appear to specify the number of sampling steps used during training, which also undermines the reproducibility of the method.

3. Although migrating the implementation to established inference frameworks is a commendable step, the paper lacks a detailed justification for the chosen configuration, specifically, the chain length of 3 and the decision to omit the draft tree. Moreover, we observe that even with a batch size of 1 (contrary to the large-batch setting emphasized in the paper), the reported speedup on H100 using *sglang* falls significantly short of earlier results. This discrepancy suggests that the earlier performance gains may have limited applicability in real-world scenarios. As such, the proposed method has yet to convincingly demonstrate its suitability for large-scale deployment, rather than being merely a toy-scale solution.

4. The authors report utilizing approximately 5,000 A100 GPU hours to train the 70B model. However, based on the parameter partitioning strategy described, where full backpropagation is not required, the computational workload appears to be only slightly greater than that of inference and significantly less than that of full model training. The reported training duration thus raises questions and calls for further clarification.

---

> ### Author Rebuttal · Authors · 2025-07-30
>
> Thank you for your time and effort in reviewing our manuscript. We sincerely appreciate your valuable comments and suggestions. Below, we provide detailed responses to your concerns.
>
> > Q1. While the paper mentions the fusion of three hidden layers (denoted as $l$, $m$, and $h$), it provides no details on how these specific layers are selected.
>
>
> A1. We select the 3rd, $n//2$-th, and $(n{-}3)$-th layers, where $n$ is the total number of layers. The table below presents experimental results on different layer selections. As long as the final layer is not used and the features are fused, the performance improves significantly, while the specific choice of layers has limited impact. Furthermore, increasing the number of layers beyond three does not bring additional gains. Therefore, we opt for the $3$-rd, $n//2$-th, and $(n{-}3)$-th layers in our design. The evaluation is conducted on the MT-Bench dataset, with LLaMA-Instruct 3.1 8B as the target model.
>
> | Layers | Accept Length    | Speedup    |
> |------------|------|------|
> | $n$      | 5.37 | 3.82x |
> | $n-1$    | 5.65 | 4.00x |
> | $n-2$    | 5.76 | 4.06× |
> | $n-3$    | 5.76 | 4.05x |
> | $n-3$, $n−3$    | 5.93 | 4.26x |
> | $n-3$, $n//2$    | 6.02 | 4.31x |
> | $3$, $n//2$, $n−3$   | 6.13 | 4.40× |
> | $1$, $n//2$, $n−1$   | 6.11 | 4.38× |
> | $5$, $n//2$, $n−5$   | 6.09 | 4.37× |
> | $3$, $n//2-5$, $n//2+5$, $n−3$   | 6.11 | 4.37× |
> | $3$, $10$, $n//2$, $n-10$, $n−3$   | 6.15 | 4.39× |
>
> > Q2. The authors do not appear to specify the number of sampling steps used during training, which also undermines the reproducibility of the method.
>
>
>
> A2. Thank you for the reminder. We used 7 sampling steps during training, and we will clarify this detail in the next revision.
>
> > Q3. Although migrating the implementation to established inference frameworks is a commendable step, the paper lacks a detailed justification for the chosen configuration, specifically, the chain length of 3 and the decision to omit the draft tree.
>
>
>
>
> A3. Speculative sampling utilizes the redundant computational capacity of GPUs. However, under the high batch size setting targeted by SGLang, such redundancy becomes limited. Consequently, we disable the computationally intensive tree-based drafting. Moreover, longer speculative chains require more resources, and our experiments show that a chain length of 3 achieves the best performance across most batch sizes. Therefore, we adopt this configuration in all our experiments.
>
> > Q4. Moreover, we observe that even with a batch size of 1 (contrary to the large-batch setting emphasized in the paper), the reported speedup on H100 using sglang falls significantly short of earlier results. This discrepancy suggests that the earlier performance gains may have limited applicability in real-world scenarios. As such, the proposed method has yet to convincingly demonstrate its suitability for large-scale deployment, rather than being merely a toy-scale solution.
>
> A4. On fixed hardware, all acceleration techniques and engineering optimizations aim to approach the hardware’s performance limits. Naturally, SGLang—already equipped with many optimizations—is more difficult to accelerate further compared to an unoptimized Huggingface setup. It is widely acknowledged in the community that production environments are harder to accelerate than experimental ones, and this applies to all acceleration methods. Accordingly, even modest improvements in production settings directly translate to cost savings.
>
> SGLang exemplifies the type of large-scale deployment scenario you referred to. Our experiments show that EAGLE-3 performs effectively in this environment, achieving a 2–3× reduction in latency compared to vanilla inference on SGLang. This highlights that our method extends well beyond toy-scale applications. Moreover, EAGLE-3 has been **officially integrated into large-scale inference frameworks such as SGLang and vLLM**, further demonstrating its practicality and scalability in real-world production systems.
>
> > Q5. However, based on the parameter partitioning strategy described, where full backpropagation is not required, the computational workload appears to be only slightly greater than that of inference and significantly less than that of full model training. The reported training duration thus raises questions and calls for further clarification.
>
>
> A5. Although EAGLE-3 uses only a single layer, it requires 7 sampling-training cycles, making the overall training cost relatively high (roughly equivalent to training 7 layers in a 70B model with 80 layers in total). Moreover, since our focus is on inference efficiency, we did not apply specific optimizations to the training process. In fact, organizations that require the deployment of 70B models typically have access to abundant computational resources, making this level of overhead acceptable.

---

> ### Author Response · Authors · 2025-08-05
>
> Dear Reviewer o9ZR,
>
> Thank you once again for your thoughtful and constructive feedback on our paper. We truly appreciate the time and effort you’ve dedicated to the review process.
>
> As a gentle reminder, the discussion period will conclude on August 6 (AoE). We’ve provided detailed responses to your comments in our rebuttal and sincerely hope they address your concerns.
>
> If you have any further questions or additional suggestions, we’d be more than happy to address them.
>
> Thank you again for your support and engagement!
>
> Warm regards,
>
> Authors of Submission 13079

---

> ### Comment · Reviewer_o9ZR · 2025-08-05
>
> Thank you to the authors for their detailed responses to my questions. I believe the main weaknesses of the manuscript lie in its casual treatment of key hyperparameters and the performance degradation observed when integrated with the inference framework. That said, the authors have addressed my concerns well, particularly the first one, through additional experiments and clarifications.
>
> Regarding the second issue, while the authors have provided a plausible explanation for the performance drop, the response does not fully resolve the underlying concern.
>
> Overall, based on the revisions and clarifications provided, I will raise my review score. I wish the authors all the best.

---

> > ### Author Response · Authors · 2025-08-05
> >
> > Dear Reviewer o9ZR,
> >
> > Thank you so much for your support. We really appreciate it.
> >
> > Best,
> > Authors

---

### Decision · Program_Chairs · 2025-09-17

**Decision:**

Accept (poster)

**Comment:**

The paper presents a practically meaningful improvement to speculative decoding with consistent empirical gains, viable integration in production frameworks, and informative ablations and data scaling analyses. The main reservations—limited theoretical analysis, narrower architectural breadth, smaller MoE and production-layer speedups, and initial reproducibility gaps—reduce confidence for a spotlight/oral.